# The Measurement of Love: Psychometric Properties and Preliminary Findings of the Short Love Scale (SLS-12) in a Polish Sample

**DOI:** 10.3390/ijerph192013269

**Published:** 2022-10-14

**Authors:** Alicja Kozakiewicz, Zbigniew Izdebski, Joanna Mazur

**Affiliations:** 1Department of Humanization of Health Care and Sexology, Collegium Medicum, University of Zielona Góra, 65-046 Zielona Góra, Poland; 2Department of Biomedical Aspects of Development and Sexology, Faculty of Education, Warsaw University, 00-001 Warsaw, Poland

**Keywords:** love, romantic relationships, validation, psychometric properties, COVID-19

## Abstract

There has been a perceived need for the development of instruments to assess relationship quality and love. The aim of this study was to describe the psychometric properties of the Short Love Scale (SLS-12) and to compare different measurement models. Data were collected using the CAWI (Computer-Assisted Web Interview) in Poland in early 2022 (18–60 years old; *n* = 941) among individuals living in formal or informal monogamous relationships. Both exploratory (EFA) and confirmatory factor (CFA) analyses were performed on two different subsamples obtained through random splitting of the full datafile. There were 12 items considered, which form three dimensions in accordance with the theoretical basis. CFA analysis confirmed good psychometric properties of the three-factor SLS-12 model based on EFA solution: χ^2^(47) = 146.802 (*p* ˂ 0.001); χ^2^/df = 3.123; CFI = 0.981, TLI = 0.973, RMSEA = 0.067 (90% CI 0.055–0.080), GFI = 0.952, AGFI = 0.921. SLS-12 subindices varied according to gender and relationship status. Based on ROC curve method, it may be assumed that scores on the SLS-12 ranging from 12 to 44 indicate a poor relationship, scores ranging from 45 to 52 a moderately good relationship, and scores of 53 to 60 a very good relationship. Improving and increasing the range of measures of love available to researchers remains an important task in supporting the progress of this area of research. Further research should be conducted among people of all ages living in both formal and informal relationships using the Short Love Scale-12 outlined in this paper.

## 1. Introduction

Love, one of the motivations to engage in romantic relationships, can have many different meanings for each person. In an effort to discover how intense or diverse love is, researchers are confronted with difficulties in measuring this phenomenon. Similar challenges are posed when evaluating intelligence, personality, anxiety, and other abstract constructs traditionally assessed by psychometrics.

Romantic relationships and experiences are an important source of emotional connection and contribute to the development of a positive self-image as well as greater social integration [1,2]. Individuals in happy relationships report higher levels of subjective well-being than those in unhappy relationships, regardless of relationship status [3]. It has also been found that people who are married or live in a marriage-like relationship are happier and enjoy better mental and physical health than those who are single [4,5,6]. One study has found that compared to those who were “very happy” in their marriage, those who were “not very happy” were more than twice as likely to report poorer health and almost 40% more likely to die over the follow-up period. Those who were not very happy in their marriage also had equal or worse health and mortality risk compared to those who were never married, divorced, separated, or widowed [7].

Efforts by psychologists to understand romantic relationships and love have increased considerably in recent decades and various theories and models of love continue to be considered [8]. Although love has historically been considered a one-dimensional concept, the dominant understanding in today’s literature is that it exhibits a multidimensional structure. Due to the complex nature of love, it may be that the level of precision desired in the field may never be achieved, and researchers and practitioners are advised to examine the definition of the construct of love and the psychometric properties of tools in selecting measures of love appropriate for their purposes [9]. Furthermore, some of the authors identify ways in which theories and models offer conflicting predictions about the origins of romantic passion, while proposing a more comprehensive model that integrates complex perspectives [10].

The diversity of theories in literature is significant, and romantic love appears to be an nearly universal phenomenon, occurring in every era. The authors will mention only some of the theories, while realizing their multiplicity. One theory that has been proposed is the “Love Color Theory” by Lee [11]. In his theory, Lee focused on the true shape of love and proposed an original and unique typology of love. Lee identified six love styles, which he placed within a closed circle and triangles: three primary types of love styles (Eros, Ludus, and Storge) and three secondary ones (Mania, Pragma, and Agape).

In the original triangular theory of love, Sternberg [12,13] defined love as consisting of three elements: intimacy, passion, and commitment. Passion is associated with the experience of desire, the manifestations of which include kissing, caressing, and the sexual act. Intimacy is viewed as positive feelings experienced in the presence of and because of a partner, through sharing experiences, receiving emotional support, exchanging information of an intimate nature and building closeness. The commitment component refers to maintaining the relationship, that is, taking various actions to sustain the relationship and make it satisfying.

The Sternberg Triangular Love Scale (STLS) contains 45 items, with an equal number of items measuring intimacy, passion, and decisions/commitment. Each of these items is rated on a Likert-type scale of 1–9, where 1 = “not at all”, 5 = “moderately”, and 9 = “extremely”. There are no labels for the scores in between those indicated above. The Triangular Love Scale still has a blank space in all 45 statements to be filled with the name of the same loved one for all the questionnaire items [14]. 

Based on Sternberg’s [12] theory, Wojciszke [15] proposed that a romantic relationship can be divided into six phases: falling in love (passion only), romantic beginning (passion and intimacy), complete love (passion, intimacy, commitment), companionate love (intimacy and commitment but no passion), empty love (commitment only), and dissolution (commitment withdrawn). He measured the intensity of each component of love in each phase, and his results were consistent with Sternberg’s [12] theoretical assumptions.

With the growing amount of literature on the triangular theory of love, various studies have examined the psychometric properties of the TLS [16,17]. Sorokowski [18], meanwhile, examined the psychometric properties of this scale in 25 countries. Moreover, a closer analysis of the studies that used the Sternberg Triangular Love Scale revealed that the samples used therein consisted mainly of married individuals [19], adolescents [17,20] or college students [21,22,23]. These studies were conducted only with individuals who were emotionally attached to their partner.

Soloski, Pavkov, Sweeney, and Wetchler [24] modified the original form of Sternberg’s Triangular Love Scale. The revised form allowed the participants to consider their general and realistic expectations of love in a marriage. The items were modified to allow for an overall assessment of expectations. Specifically, this study found that lower levels of inter-parental conflict were significantly related to the higher levels of love an individual expresses in their relationship as well as higher levels of marital expectations of love. Furthermore, it is believed that due to the three-factor structure of the STLS and high levels of reliability, the scales can be used to determine the level of love and love expectations of young adults [25].

Lemieux and Hale [26,27] developed an alternative tool with only 19 items that captured these three dimensions. Although to date this measure has only been used in samples of college students or married individuals, its conciseness and simplicity makes it a suitable measure for assessing romantic relationships in adolescents as well, as tested by Overbeek et al. [17].

Furthermore, the Passionate Love Scale is a 30-item scale (often shrunk into a 15-item measure) designed to measure an individual’s level of passionate love toward another person. Each item has a blank space and respondents are instructed to fill it in with the name of their partner. Each item is answered on a 9-point scale ranging from 1 (not at all true) to 9 (definitely true) [28]. It is worth pointing out, however, that none of the items in this scale relates directly to sex life.

It should be noted that in the literature, both Polish and foreign, there are no scales with a fewer number of items that could be used in a multi-subject study where questionnaire length limitation is relevant. The scale created in Poland and presented in this paper consists of 12 items. Demonstrating the psychometric properties of the presented scale will contribute to the diversification of research focusing on love and romantic relationships. Proposed an even smaller number of items than before, which will allow the use of scale in a more extensive context. Some of the questions presented were previously tested in Izdebski’s works [29,30], but no study has attempted to treat them as a scale. After preliminary analyses of the results of previous surveys, new questions have been added and the response category has been expanded from a range of “never”–“constantly” to one of “very much fits the description of my relationship”–“does not fit the description of my relationship at all”.

Linguistic and cultural adjustments often generate difficulties in translating the original questions and obtaining sufficient psychometric properties. Scales that were developed from the beginning in a given social and cultural context may be more suitable. In addition, as we pointed out, the construction of SLS-12 is a continuation of our own research, since a number of items has been derived from earlier studies. 

It was decided that two items will extend to include questions directly related to sex life, which were not included in the previous love scales, while according to the tool’s authors they are an important aspect of understanding love. It is also worth noting that sexuality results from the interactions taking place between the individual and the surrounding social structure, and its full development is necessary for individual, interpersonal and social well-being while representing an important, although not the only aspect of a romantic love relationships. 

In addition, Sternberg’s original scale can be used for a variety of attachment objects, such as siblings or parents, while the proposed scale, due to the sex life component, is narrowed down only to close romantic relationships or imagined romantic relationships. This constraint allows for the inclusion of the aforementioned area of sexual life and communication in this area. 

Sternberg’s scale provided an important reference point for theoretical considerations when constructing the scale, given the fact that there are not many thoroughly researched tools. None of Sternebrg’s scale questions were adapted in identical wording, although the concept of multidimensional love was close to the idea of creating a scale by itself. Nevertheless, the authors’ objective is not to provide a critique of it, but to propose a tool that reflects the needs of psychometric measurement.

Given the need for research into romantic relationships in Poland, the main aim of this study was to develop the Short Love Scale-12 (SLS-12), assess the psychometric properties of the scale and to compare different measurement models. The authors of this study hope to identify a reliable, valid, and feasible tool for research purposes.

## 2. Materials and Methods

### 2.1. Participants and Procedure 

The survey was conducted as part of a larger project on the humanization of medicine from 2–17 March 2022. The sample size was 2050 (*n* = 2050) and it included both adult patients and legal caretakers of patients who received medical care in 2020 and/or 2021. This study used a self-administered online survey (CAWI) technique registered with a research panel provided by Research Collective Company. An individual link to the electronic survey form was sent with the survey invitation. Each respondent was double-checked for consistency in their answers to the questions about demographic information.

The survey questionnaire contained questions regarding patient’s evaluations of various aspects of their relationships with health care professional, as well as questions about certain aspects of their own lives, and the impact of the COVID-19 pandemic on their evaluations. The respondents provided answers to close-ended questions, mainly on nominal or ordinal scales. The questionnaire consisted of 296 variables. However, only a small part was used in this study. The average time of completion was 28.8 min, and the median value reached 23.9 min. Both were calculated considering only fully completed questionnaires.

For the purpose of this study, a group of *n* = 941 (no missing data for key variables) participants was selected. The inclusion criteria were ≤60 years of age and in a monogamous relationship for at least two months. The group under evaluation was gender-balanced (45.6% male, 54.4% female). The age of the respondents ranged from 18–60 years. The sample was regionally diverse, covering all major administrative units in Poland (regions), and the percentage of urban and rural residents was 53.8% or 46.2%, corresponding to data from the Demographic Yearbook [31]. In the selected group, 69.4% of the respondents reported living in a formal relationship and 30.6% in an informal one. In addition, 94.4% declared being in a relationship for more than a year and the remaining 5.6% for less than a year. The median duration of the relationship was 12 years. A total of three data sets were used for the statistical analyses: the whole dataset (*n* = 941); and two subsets of the whole dataset obtained by a random permutation of the respondents’ order, which was then split into two halves (the first used for EFA, *n* = 471, the second used for CFA, *n* = 470). The descriptive statistics of the three datasets are presented in Table 1 and show that the characteristics of the three data sets are very similar. 

### 2.2. Tools 

The SLS-12 scale was developed by the project team based on theories of love. An earlier self-report study analyzed a set of 19 statements, six of which were categorized in the national report as a consistent scale for assessing romantic relationship quality. After analyzing past research results and reviewing the literature, it was decided to expand the measurement scale. Each item is presented on a five-point Likert scale, where 5 is very much fits the description of my relationship, while 1 is not at all fits the description of my relationship. The maximum score = 60 and the minimum score = 12. The overall index takes a range of 12–60. For both the overall index and the subscales, high scores indicate a stronger relationship.

A question about intention to reconnect with the same person was included in the analyses, which helped us to categorize the overall SLS-12 index and to distinguish the critical point on this scale, below which we can talk about a crisis in the relationship. In the analyzed group of 941 respondents, 72 people (7.7%) refused to answer this question. In the remaining group of 869 respondents, the majority would definitely like to be (54.1%) or would like to be (30.1%) in a relationship with the same person. The percentage of negative evaluations was 15.8% (11.2% rather not and 4.6% definitely not). 

The Experiences in Close Relationships-Revised Scale (ECR-RS) was used as a validation tool. It is a 9-item self-report instrument designed to assess attachment patterns in various close relationships. The test–retest reliability (over 30 days) of the individual scales are approximately 0.65 for the domain of romantic relationships (including individuals who experienced breakups during the 30-day period) and 0.80 in the parental domain. The items of the ECR-RS are as follows: It helps to turn to people in times of need; I usually discuss my problems and concerns with others; I talk things over with people; I find it easy to depend on others; I don’t feel comfortable opening up to others; I prefer not to show others how I feel deep down; I often worry that other people do not really care for me; I’m afraid that other people may abandon me; I worry that others won’t care about me as much as I care about them [32].

For the purposes of this study, the author’s consent was obtained and the Polish translation by M. Marszal was used. The Polish translation was provided by the tool’s author at the request of the research team. The scale is a self-referential tool and consists of two dimensions—anxiety and avoidance. The same nine items are used to assess attachment styles in relation to four targets (i.e., mother, father, romantic partner, and best friend). However, for the purposes of our study, the assessment was measured only in relation to the romantic partner. 

### 2.3. Ethical Consideration 

The questionnaire could have been left with incomplete answers at any time, without giving any reason and without any consequences. The respondents proceeded to fill out the questionnaire voluntarily, having full information about what participating in the research meant to them. At the same time, consent to participate in the survey was given by choosing yes or no answers on the computer screen, due to the online nature of the survey. The procedure and tools used in this research project were approved by the Research Ethics Committee at the Faculty of Pedagogy, University of Warsaw No. 2021/8.

### 2.4. Statistical Analyses 

The statistical analysis included a thorough assessment of the reliability and validity of the SLS-12 and the distribution of responses in the overall index and three sub-indexes. An attempt was also made to determine the cutoff point for identifying stronger and weaker relationships. 

Frequency distributions for the items and summary measures were described using measures of average and dispersion values and the frequency of outliers. An item is considered to exhibit a floor or ceiling effect when a large percentage of respondents are at the outliers of the scale [33]. Effects of up to 15% were considered acceptable [34]. Skewness and kurtosis were estimated to confirm the normality of the data using item analysis and verified them by the multivariate normality test (*n* = 470) using AMOS. To estimate multivariate skewness and kurtosis Mardia’s coefficient was calculated using SPSS macro [35].

The Cronbach’s alpha coefficient was used to estimate the internal consistency of the data in each scale. In general, values for Cronbach’s alpha above 0.70 are considered to indicate a reliable set of items. [36]. 

Additionally, as a measure of convergent validity correlations between subscales were counted using Spearman’s rho correlation. Convergent validity is generally considered adequate if a correlation with an instrument measuring the same construct is >0.50. It is assumed that weak correlation is <0.3, moderate 0.3–0.5 and high correlation >0.5. 

The main method for evaluating the structure of the SLS-12 scale was confirmatory factor analysis (CFA). The use of exploratory factor analysis (EFA) results was also planned. For the EFA and CFA analyses, the group was divided into two equal groups (*n* = 471 and *n* = 470) using random splitting of the full datafile. This allowed comparison of alternative models based on the 12 statements of the SLS-12 scale. EFA results were placed as supplementary electronic material. 

An EFA was performed using the PCA (principal component analysis) as extraction method and Promax with Kaiser normalization on the 12 items with the critical factor load value below 0.4, due to theoretical basis of the original tool proving the possibility of the correlation of the factors [37]. When conducting the EFA analysis, various methods of factor extraction were adopted: eigenvalues over 1 and pre-assumed number of factors.

Then, a confirmatory factor analyses (CFA) was conducted for four models—one factor based on EFA solution, three factor theory driven and three and four factor based on exploratory factor analyses (EFA) results. To perform a CFA, the maximum likelihood estimation method together with 5000-bootstrap samples were used due to the violation of the multivariate normality assumption. Bootstrapping is a robust procedure for dealing with non-normality in multivariate data [38,39,40]. It was also conducted in order to generate accurate estimations of standard errors with accompanying confidence intervals (bias-corrected at the 95% confidence level) and *p*-values. Looking for the optimal solution, modification indices (MI) were used, correlating the residuals within each of the three factors if the MI was >10. The following model fit indices were reported as results: CMIN/DF, comparative fit indices (CFI), goodness-of-fit index (GFI), adjusted goodness-of-fit index (AGFI), and root mean square error of approximation (RMSEA) were used to evaluate the model fitness. The AGFI, GFI, CFI, Tucker–Lewis’s index (TLI), and normed fit index (NFI) ≥ 0.90 indicate a good and adequate fit of the model to the data [41]. In the confirmatory factor analysis, RMSEA values < 0.08 were considered significant [42,43]. For CMIN/DF, a value lower than 5 indicates a reasonable fit, although a value lower than 3 is preferable [44].

The non-parametric Mann–Whitney U test was used to examine differences in the scores of the scale and its factors analyzed by gender and the nature of the relationship.

A receiver operator characteristic (ROC) curve analysis was used to distinguish the three categories of the SLS-12 general index scale. The response to the question about the intention to reconnect with the same person was used as an external criterion for assessing the relationship with the partner. Optimal score thresholds were determined using Youden’s J-index, and calculated according to the formula sensitivity + specificity − 1. ROC curves were further evaluated with the AUC (area under curve) index. Two ROC curves assessed the ability of the SLS-12 scale to diagnose moderately good and very good relationships.

Statistical Package for the Social Sciences (SPSS) 27.0 and Amos 26.0 (IBM, Armonk, NY, USA) were used for data analysis. 

## 3. Results

### 3.1. Descriptive Statistics

Table 2 shows the basic descriptive statistics for the analyzed items and the floor and ceiling effects. It can be observed that the distribution of the analyzed items is significantly different from the normal distribution. All the items show a left skewness with a maximum absolute value of 1.349. The kurtosis coefficients range from −0.230 to 1.823. The analysis of the floor and ceiling effects shows a low proportion of extremely negative responses and a very high proportion of extremely positive responses. The proportion of extremely positive responses ranged from 30.3% to 53.9% and was highest in items 5, 6, 8, and 9. The proportion of extremely negative responses ranged from 1.5% to 6.4% and was lowest in items 9, 5, and 11. Mardia’s coefficient yielded a value of 293.776 for multivariate kurtosis and a value of 32.913 for multivariate skewness, both with *p* < 0.01, indicating non-normality. 

### 3.2. Exploratory and Confirmatory Factor Analysis 

One-factor EFA model was based on criterion eigenvalues over 1 as suggested solution (Appendix A). Appendix A presented four-factor model pre-assumed in EFA. Finally, using EFA it was reached a 12-item and 3-factor version (Appendix A). For this final solution, the first factor explains 70.59% of the variance, while the cumulative variance values explained, respectively, by the second and third factors are 77.20% and 80.52%

Confirmatory factor analyses were employed to investigate to what extent the four different models fit the data. The 1-factor model assumes that all items of the general version load on one single factor (Appendix A). The theory driven model assumes three correlated factors: passion, intimacy and commitment (Figure 1). The three-factor SLS-12 model was based on EFA results (Figure 2), and also the four-factor SLS-12 model was based on EFA results (Appendix A). Table 3 shows the fit indices of the four models. The 1-factor model does not fit so well, but the other models fit well to the data well. Figure 1 shows the theory driven model. After analyzing the fit quality of this model, the CMIN/DF value was shown to be 3.142/46. The absolute fit and incremental fit parameters of the model indicated insufficient goodness of fit, as the values were all >0.90 (CFI = 0.981, TLI = 0.973, NFI = 0.973, RFI = 0.961, incremental fit index (IFI) = 0.981, GFI = 0.953, AGFI = 0.921). The RMSEA and RMR values were 0.068 (90% CI 0.055–0.080) and 0.019. 

Due to the fact that factors were highly correlated with each other in theory driven SLS-12 model, the authors assumed a three and four factor model based on EFA results. In a three-factor confirmatory model CMIN/DF value was shown to be 3.123/47, indicating an acceptable result. The absolute fit and incremental fit parameters of the model indicated a good fit, as the values were all >0.90 (CFI = 0.981, TLI = 0.981, NFI = 0.972, RFI = 0.961, incremental fit index (IFI) = 0.981, GFI = 0.952, AGFI = 0.921). The RMSEA and RMR values were 0.067 (90% CI 0.055–0.080) and 0.018 (Table 3). 

For the three-factor SLS-12 model, which shows the best fit, the values of the standardized regression coefficients ranged from 0.80 to 0.90. Thus, all the factor loadings were above 0.50 and significant (*p* < 0.01). (Table 4). The covariance between factor 1 and 2 was 0.87, *p* < 0.01; between factor 1 and 3 0.77, *p* < 0.01; and between factor 2 and 3 0.92, *p* < 0.01 (Figure 2). All models (except one-factor) demonstrate satisfactory fit models, nevertheless the three-factor model (Figure 2) shows the best fit indices. The theory driven model exhibited a high standardized coefficient between the commitment and intimacy factors (Appendix A). Second-order models found to be no improved due to the high standardized coefficient between factors. For theory driven SLS-12 model the results was CMIN/DF = 3.142/46, RMSEA = 0.068 and CMIN/DF = 3.123/47, RMSEA = 0.067 for three-factor SLS-12 model.

The SLS-12 scale consists of 12 items (Appendix B) with three subscales: 2 items, 2 items and 8 items. Four items were taken from the scales applied in the previous quantitative studies, while the remaining ones were developed through a team discussion for the purpose of the present study. Factor 1 relates directly to sex life, factor 2 speaks of longing and affectionate gestures, while factor 3 focuses on mutual respect, support, the capacity to resolve disagreements and a sense of security in a relationship.

### 3.3. Reliability and Validity 

Descriptive statistics for the three-factors of the SLS-12 scale are presented in Table 5. Internal consistency values indicate good reliability for all three factors (Cronbach’s alpha = 0.849 for factor 1, Cronbach’s alpha = 0.826 for factor 2, and Cronbach’s alpha = 0.957 for factor 3). The total Cronbach’s α of the scale was 0.959, indicating good internal consistency of the entire scale. Moreover, the total Cronbach’s α of the ECR-RS was 0.876, indicating good internal consistency.

All subscales of SLS-12 and ECR-RS correlated at moderate to strong levels (Table 6). A moderate negative correlation (r = −0.535; *p* < 0.01) was observed between factor 1 and avoidance. There was a strong correlation between factor 2 and avoidance (r = −0.642; *p* < 0.01), factor 3 and avoidance (r = −0.721; *p* < 0.01), factor 3 and anxiety (r = −0.561; *p* < 0.01). A moderate correlation was observed between factor 1 and anxiety (r = −0.364; *p* < 0.01) and factor 2 and anxiety (r = −0.406; *p* < 0.01). The SLS-12 total scores showed a very strong negative correlation with the ECR-RS total scores (r = −0.713; *p* < 0.01).

The results of these two scales are correlated with each other, the negative correlation is related to the fact that high ECR-RS scores indicate a negative assessment of attachment in a close relationship. Considering the aforementioned results, this assessment can be said to have high convergent validity.

### 3.4. SLS-12 Distribution by Gender and Relationship Status 

When considering the construct of love, statistically significant differences were observed in the scores for gender differences and the nature of the relationship. A Mann–Whitney U test was calculated to determine if there were any significant differences in responses to individual items and between subscales among men and women, and those in formal and informal relationships. Statistically significant differences were found between women (N = 512) and men (*n* = 429) for items 6 (M_Men_ = 4.29, M_Women_ = 4.12, Z = −2.307, *p* = 0.021), 7 (M_Men_ = 4.11; M_Women_ = 4.22; Z = −2.431; *p* = 0.015), 8 (M_Men_ = 4.20; M_Women_ = 4.30; Z = −2.705; *p* = 0.007), 11 (M_Men_ = 4.04; M_Women_ = 4.13; Z = −2.013; *p* = 0.044). Men scored higher for all items. Gender differences were also significant for the Factor 3 (M_Men_ = 33.96; M_Women_ = 132.76; Z = −2.106; *p* = 0.035).

Formal (N = 653) and informal (N = 228) relationships showed differences for items 1 (M_Formal_ = 3.85; M_Informal_ = 4.12; Z = −4.119; *p* < 0.001), 3 (M_Formal_ = 3.64; M_Informal_ = 3.85; Z = −3.105, *p* = 0.002), 4 (M_Formal_ = 3.89; M_Informal_ = 4.09; Z = −2.959, *p* = 0.003), 5 (M_Formal_ = 4.25; M_Informal_ = 4.44; Z = −3.401; *p* < 0.001). Differences in formal and informal relationships proved to be significant for the Factor 1 (M_Formal_ = 7.41; M_Informal_ = 7.72; Z = −2.508; *p* = 0.012) and Factor 2 (M_Formal_ = 7.74; M_Informal_ = 8.21; Z = −3.705; *p* = 0.001). Statistically higher scores were achieved by people in informal relationships for all items.

### 3.5. ROC Curves 

Figure 3 illustrates the two ROC curves facilitating the classification of the overall SLS-12 index. Adopting the criterion of intention to reconnect with the same person (yes or definitely yes) yielded very good curve properties (AUC = 0.914) and an optimal split point corresponding to a score of 45. Assuming the more stringent external criterion of definitely confirming the intention to reconnect with the same person yielded slightly worse but satisfactory curve properties (AUC = 0.858) and a split point corresponding to a score of 52 points. Thus, it can be conventionally assumed that the scores on the SLS-12 general scale ranging from 12 to 44 indicate a poor-quality relationship, sores ranging from 45 to 52 a moderately good relationship, and scores of 53 to 60 a very good relationship. In the sample of 941 people, 28.1%, 29.4%, and 42.5% of the respondents were in each group, respectively.

## 4. Discussion 

Assessing the quality of romantic relationships with the SLS-12 may be an important step towards better measuring the self-referential relationship an individual is currently in. Intimate relationships become increasingly important as individuals move from dating during adolescence, to cohabiting as they enter adulthood, and finally marriage in adulthood [45]. Furthermore, there is ample evidence that higher marital quality has been associated with better health, including lower mortality risk and lower cardiovascular reactivity during marital conflict [46]. Additionally, the results of the study indicate a significant variation in adolescents’ experiences of romantic relationships and point to the developmental importance of these experiences for short- and long-term well-being [47].

The SLS-12 is a 12-item scale that can be used in both individual and group studies of people in formal or informal relationships. Such a short and easy to use tool is particularly important in projects involving multithreaded questionnaires that require a short completion time. However, given that short scales are usually developed for research and group statistics rather than for diagnostic purposes at the individual level [48], the authors recommend that this type of scale be used in population-based research. The overall score is the sum of the indices from the three dimensions, which can also determine a person’s “love profile”.

Overall, the results of this study indicate that the SLS-12 is a highly reliable instrument with good convergent validity and adequate construct validity. Construct validity was demonstrated by statistically significant moderate, strong, and very strong negative correlation values between the three subscales of the SLS-12, and the two subscales of the ECR-RS. 

The results indicate that the distribution of individual scale scores corresponding to the factors of the SLS-12 scale differed by gender and relationship type. In Factor 3, related to mutual respect and support, men had statistically higher scores, while in Factors 1 and 2, focusing on sex life, mutual longing and showing affection, scores were statistically higher in those in informal relationships. It is noteworthy that initial high levels of passion often end in habituation [12]. Acker and Davis [49] found that passion declines over time in long-term relationships. In the future, the SLS-12 scale may be correlated with broader assessments of sex life. 

The strength of this study is that it attempts to categorize the overall SLS-12 index using an external criterion. However, it is worth recalling, in the context of the ROC analysis, that 7.7% of the subjects did not want to answer the question regarding their intention to be involved with a romantic partner, which may indicate problems in their relationship. A disadvantage of the SLS-12 scale is left-skewed distribution with a clear dominance of positive ratings. This is due to the peculiarity of the phenomenon studied, i.e., the common good evaluation of the relationship with a partner. However, future research with SLS-12 should pay more attention to those who evaluate their relationships less positively, which was the case in every sixth respondent in our study.

It should also be noted that in Izdebski’s [30] study of the relationship rating in the current relationship and the response to the question on divorce intentions, among those who are considering divorce, the relationship rating drops to 24.5 points, compared to 32.4 for those not thinking about divorce. With very positive responses to questions about, among other things, mutual understanding, being happy with each other, showing tenderness, being tense, and sexual attraction, thoughts of divorce among respondents hardly appeared. 

In this and in earlier works [18,50,51], correlations between subscales were highly significant. One possible reason is simply that dimensions of love tend to occur together in most, though certainly not all, relationships based on love. In the early stages of a successful relationship especially, people may idealize their partners [52]. Romantic idealization, in its most global sense, is “the tendency to describe the relationship [and the partner] in unrealistically positive terms” [53] p. 7. Accordingly, subjects may significantly agree with the (positive) statements in love scales. Moreover, to avoid cognitive dissonance [54], people want to affirm that they have chosen their partner appropriately, so they tend to indicate somewhat positive responses. These two factors may further explain the high ceiling effect in all the analyzed items. 

Specifically, studies have shown that friendship and romantic relationships are characterized by a multidimensional nature that implies the presence of similar both positive (intimacy, support, aid, appreciation and admiration, specialness, nurturance and affection, togetherness or reliable alliance, exhilaration and companion- ship) and negative aspects (painfulness, conflict and negative interactions) [55]. The multidimensionality of the love construct can be measured in sub-scales, but the authors also allow it to be measured as a single factor. As the three SLS-12 short form subscales each had acceptable internal consistency and were highly correlated with each other (Figure 2). Despite the fact that there is no sufficient fit in one-factor model, it is possible to use the summary result based on EFA results.

The proposed three-factor model based on EFA results is not in opposition to the original Sternberg model. However, the high standardized coefficient between factors encouraged us to search for other solutions. The proposed optimal model integrates intimacy and commitment, correlated at 1.00 on theory driven model (Appendix A). When questions directly related to sex life (not previously included) were added to the passion dimension, it proved better to make it a separate factor. The third factor that emerged expresses other aspects of love. 

In order to vary the possibility of bias, participants were not informed in advance about the nature of the triangular or any other theory, and the overarching goal of the survey was to study the humanization of medicine, not the quality of relationships. In addition, to reduce the total survey error, an online format was used which eliminated the presence of the interviewer. The respondents were assured that their answers would not be linked to them and were not asked to disclose any sensitive information directly to the interviewer. 

In the future, it may be advisable to consider behavioral or even psychophysiological measures that correspond to the aspects of love and relationship quality.

### Limitations and Future Research Directions

In assessing the strengths and weaknesses of the SLS-12 scale validation conducted, it is important to note that data were obtained via the Internet from a group of individuals enrolled in a research panel. Thus, participants had to have knowledge of computer use and Internet access. This may have eliminated some respondents who were not accustomed to working online. However, as in other validation studies of this type of scales, the oldest respondents (here aged over 60) were not included, which in itself may have led to overrepresentation of those with digital competence. 

The cross-sectional nature of the data did not allow us to assess other psychometric properties, such as predictive validity and test–retest reliability, and limited our ability to infer causal relationships. Moreover, the future research should focus on measurement invariance.

Further research is needed to compare the results obtained in the present study with more demographically diverse samples. This study was conducted among participants who had mostly been in relationships for more than one year, and the group of younger participants, under the age of 30, formed a minority. Additional analyses should focus on other subgroups, such as younger couples or those in short-term relationships, as well as attempting to evaluate the relationships of older people with very long ones. The scale can be used for further research, where one can consider expanding the two-item factors with more items.

Moreover, the authors realize that they are not citing a significant amount of works from recent years. This is because we lack current and validated tools that would measure love and relationship quality, while the theoretical foundations created years ago are still valid.

It is also worth noting that this study was conducted as part of a larger project concerning the humanization of medicine. This analysis was conducted on a group of people who had undergone medical treatment or diagnostic tests in the last two years. It can be assumed that the majority of the population has had contact with medical care during such a period, although the COVID-19 pandemic did tend to limit contact with health care units. The context of the pandemic may impact the way we infer about relationship problems, although it should have no relevance in the psychometric evaluation of the scale. The COVID-19 pandemic may have brought additional stress to relationships, altering lifestyles as well as work and family functioning. There were small moderation effects of relationship coping and conflict during the pandemic, one study found that satisfaction increased and maladaptive attributions decreased in couples with more positive functioning, and satisfaction decreased and maladaptive attributions increased in couples with lower functioning [56]. Individuals not living together had limited contact opportunities, as physical contact compounded the risk of infection, and social distancing measures and confinement to the home were recommended to prevent the spread of the SARS-CoV-2 virus. These necessary changes placed great pressure on individuals, leading to the widespread deterioration in community mental health highlighted in many publications [57,58,59]. The tool presented in this paper for examining close romantic relationships that was used in research during the COVID-19 pandemic allows for a holistic and interdisciplinary view of relationships. This topic remains highly relevant from a public health perspective and requires further investigation in the context of relationships as a protective factor among patients and others.

The aforementioned pandemic-related factors were also likely to have impact on the results of our study. Notwithstanding these limitations, it can be argued that the SLS-12 has been shown to be a promising scale for measuring different patterns of love in formal and informal relationships. Having a good tool and data from the pandemic period provides an ideal starting point for comparisons with the post-pandemic period. 

## 5. Conclusions

In conclusion, our results contribute to empirical research on the measurement of love in romantic relationships by introducing a short and psychometrically validated scale. We obtained a theoretically grounded and psychometrically robust measure of love. Importantly, the SLS-12 contains only items with high content relevance, it has been proven to have good psychometric properties, and its conciseness satisfies the need for adequate measures of love measurement suitable for time-limited research, including large-scale studies. Improving and expanding the range of measures of love available to researchers remains an important task to support the progress of this field of research. The material collected in the project on humanization of medicine provides opportunities for further analyses of the socio-demographic determinants of overall and sub-scores on the SLS-12 scale, as well as on the protective effect of a quality relationship in the face of illness or in relation to the health and psychological effects of the COVID-19 pandemic. The scale can be implemented in subsequent research projects conducted in different population groups and countries.

## Figures and Tables

**Figure 1 ijerph-19-13269-f001:**
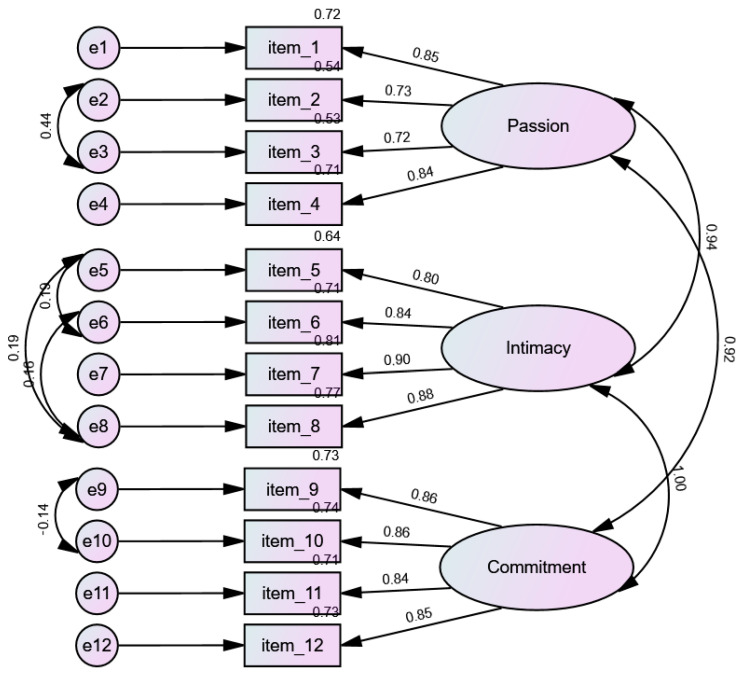
Theory driven SLS-12 model (*n* = 470).

**Figure 2 ijerph-19-13269-f002:**
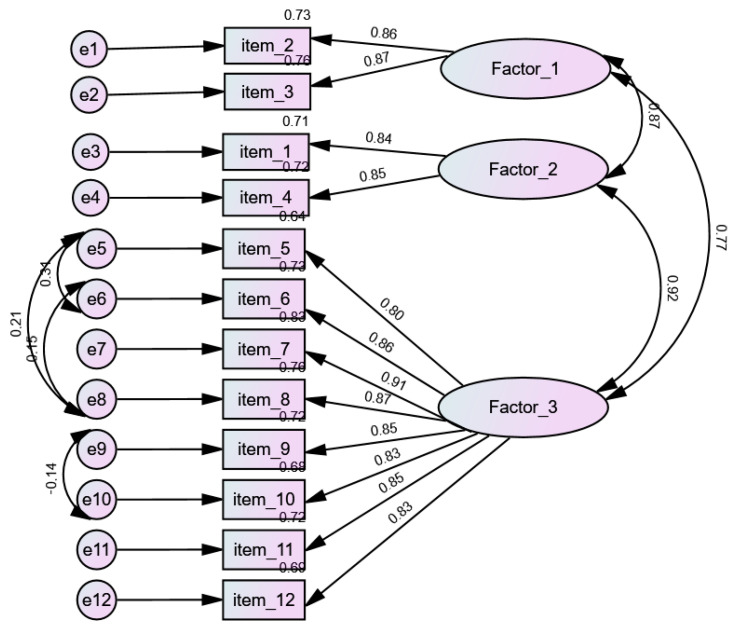
Three-factor SLS-12 model based on EFA solution (*n* = 470). Factor 1—sexual life; Factor 2—closeness; Factor 3—commitment.

**Figure 3 ijerph-19-13269-f003:**
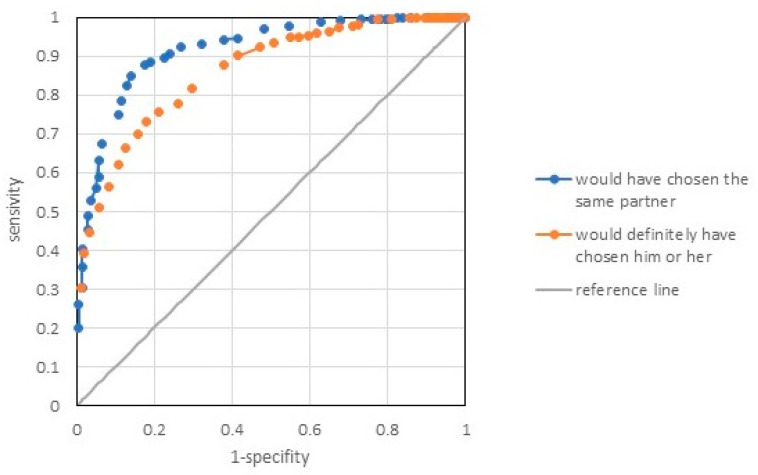
The receiving operating characteristics curve (ROC) of SLS-12 as a marker of good and very good relationships.

**Table 1 ijerph-19-13269-t001:** Sample characteristics (all data presented as percentages). The two smaller datasets were created from the whole set of data by a random permutation of the respondents’ order and then dividing the permuted dataset into two halves.

Variable	Categories	Total *n* = 941	EFA Sample *n* = 471	CFA Sample *n* = 470
Sex	Men	45.6	46.1	45.1
	Women	54.4	53.9	54.9
Age	18–29 yrs	16.2	17.6	14.7
	30–39 yrs	32.6	31.4	33.8
	40–49 yrs	27.1	26.3	27.9
	50–60	24.1	24.6	23.6
Education	Primary and vocational	31.0	32.5	29.6
	Secondary	32.5	30.4	34.7
	Higher	36.5	37.2	35.7
Place of living	Large cities	25.4	22.9	27.8
	Small towns	28.4	29.3	27.5
	Rural areas	46.2	47.8	44.7
Relationship type	Formal	69.4	69.0	69.8
	Informal	30.6	31.0	30.2
Relationship length	Less than a year	5.6	6.4	4.9
	Longer than a year	94.4	93.6	95.1
Having children	Yes	77.8	76.4	79.1
	No	22.2	23.6	20.1

EFA—exploratory factor analysis; CFA—confirmatory factor analysis.

**Table 2 ijerph-19-13269-t002:** Descriptive statistics, floor and ceiling effects for individual items (N = 941).

	M	SD	Skewness	Kurtosis	Floor Effect (%)	Ceilling Effect (%)
Item 1	3.93	1.118	−0.879	0.003	4.0	39.7
Item 2	3.79	1.165	−0.815	−0.072	6.4	33.9
Item 3	3.70	1.157	−0.681	−0.230	6.4	30.3
Item 4	3.95	1.068	−0.828	−0.019	2.9	38.9
Item 5	4.32	0.899	−1.349	1.823	1.6	53.9
Item 6	4.20	0.934	−1.145	1.018	1.7	47.7
Item 7	4.11	0.994	−1.026	0.556	2.1	44.3
Item 8	4.20	0.958	−1.180	1.026	1.9	48.5
Item 9	4.21	0.919	−1.127	0.959	1.4	47.2
Item 10	4.11	0.911	−0.993	0.883	1.6	39.5
Item 11	4.04	0.958	−0.854	0.268	1.5	38.5
Item 12	4.11	0.930	−1.002	0.832	1.8	40.8

M—mean, SD—standard deviation.

**Table 3 ijerph-19-13269-t003:** Confirmatory factor analysis of the SLS-12 scale (N = 470).

One-Factor SLS-12 Model (Based on EFA Solution)									
CMIN/DF	RMR	GFI	AGFI	CFI	NFI	RFI	IFI	TLI	RMSEA
4.671/49	0.034	0.924	0.879	0.966	0.957	0.942	0.966	0.954	0.088
Theory driven SLS-12 model									
CMIN/DF	RMR	GFI	AGFI	CFI	NFI	RFI	IFI	TLI	RMSEA
3.142/46	0.019	0.953	0.921	0.981	0.973	0.961	0.981	0.973	0.068
Three-factor SLS-12 model (based on EFA solution)									
CMIN/DF	RMR	GFI	AGFI	CFI	NFI	RFI	IFI	TLI	RMSEA
3.123/47	0.018	0.952	0.921	0.981	0.972	0.961	0.981	0.973	0.067
Four-Factor SLS-12 model (based on EFA solution)									
CMIN/DF	RMR	GFI	AGFI	CFI	NFI	RFI	IFI	TLI	RMSEA
4.026/48	0.020	0.936	0.896	0.972	0.963	0.950	0.972	0.962	0.080

**Table 4 ijerph-19-13269-t004:** Standardized regression weights for the three-factor SLS-12 model (95% CI).

			Estimate	Lower	Upper	*p*-Value
Item 1	←	Factor 2	0.843	0.796	0.883	0.000
Item 2	←	Factor 1	0.856	0.803	0.900	0.001
Item 3	←	Factor 1	0.870	0.825	0.906	0.001
Item 4	←	Factor 2	0.848	0.800	0.885	0.001
Item 5	←	Factor 3	0.802	0.749	0.847	0.001
Item 6	←	Factor 3	0.856	0.804	0.839	0.001
Item 7	←	Factor 3	0.909	0.883	0.930	0.001
Item 8	←	Factor 3	0.873	0.836	0.902	0.001
Item 9	←	Factor 3	0.846	0.797	0.883	0.001
Item 10	←	Factor 3	0.827	0.772	0.868	0.001
Item 11	←	Factor 3	0.847	0.806	0.880	0.001
Item 12	←	Factor 3	0.829	0.774	0.870	0.001

**Table 5 ijerph-19-13269-t005:** Descriptive statistics of the three factors of the SLS-12 (N = 941).

	Factor 1	Factor 2	Factor 3	SLS-Total
M (SD)	7.50 (2.16)	7.88 (2.02)	33.31 (6.58)	48.69 (10.01)
Skewness	−0.767	−0.826	−1.040	−0.939
Kurtosis	0.003	0.080	0.867	0.601
Range	8.00	8.00	32.00	48.00
Cronbach’s alpha	0.849	0.826	0.957	0.959

M—mean, SD—standard deviation.

**Table 6 ijerph-19-13269-t006:** Correlations between SLS-12 subscales and ECR-RS subscales (N = 941).

Spearman’s rho	Factor 1	Factor 2	Factor 3	SLS-Total
ECR-RS_avoidance	−0.535 **	−0.642 **	−0.721 **	−0.713 **
ECR-RS anxiety	−0.364 **	−0.406 **	−0.561 **	−0.524 **
ECR_RS-total	−0.521 **	−0.610 **	−0.737 **	−0.713 **

** *p* < 0.01.

## Data Availability

The data are owned by Warsaw University and are not to be made freely publicly available.

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
