# Peer review of "The Measurement of Love: Psychometric Properties and Preliminary Findings of the Short Love Scale (SLS-12) in a Polish Sample"

_ijerph, 2022, doi:10.3390/ijerph192013269_

Round 1

Reviewer 1 Report (New Reviewer)

The manuscript describes an initial psychometric investigation into the factor structure of the Short Love Scale-12. While the premise of the manuscript is important, there are several aspects of data analysis and study justification that prevent me from recommending its publication in its current format.  I will outline major themes and areas needing substantial revisions below. I believe that once revisions are made to the study, it would constitute an important contribution to the literature.

1.    The number of references used in the Introduction and Discussion should be significantly increased, focusing especially on those references from the last five years.

2.    It is necessary to give a direct link to the article devoted to the psychometric properties of the ECR-RS Polish version. If this article has not yet been published, provide Cronbach's alpha calculated on your sample for this tool.

3.    There are several formatting/grammatical errors in the manuscript. You should carefully review the PDF of the article before approving it.

4.    You should to detail which questions were adopted from the Sternberg scale and which ones were new. Did you adapt the language during translation?

5.    As I understand, the purpose of this study was not just a validation of existing inventory, but the developing of a new scale. Therefore, at the end of the introduction this objective should be clearly defined.

6.    There are several portions of the manuscript where not enough detail was provided. Why use promax rotation? Why did you decide to leave one, three or four factors after the PCA (theoretical justification, specific statistical procedures such as Scree plots or something else)?

7.    In fact, you use PCA for statistical analysis, although you label it as EFA. But these statistical procedures differ from each other. It would be more correct to give the "PCA" abbreviation instead of the "EFA" throughout the article.

8.    What does the "NLTS-12 scale" abbreviation mean (line 422)?

9.    For now, only descriptions of factors of your SLS-12 version are given. What are the names of the factors (new scales)? It will be appropriate to show the revised SLS-12 with the new scales and the relevant items in the separate table.

Author Response

Reviewer 2 Report (New Reviewer)

The article uses a crossvalidation method since they performed an EFA and CFA with different samples, which is excellent methodologically and it is a very interesting study. Nevertheless, there are some points that I think they should consider enhance. The CFA was conducted using the AMOS program,   and the program provides the multivariate kurtosis for the assessment of nonnormality and should report it. Also, it is necessary to report what estimator was used  in the CFA, for example, ML, ADF, Unweighted Least Squares, etc. RMSEA provides confidence intervals and should be reported.  Moreover, there are too many covariates between indicator's error terms and probably this is an indication of an intent to overfitting the model. In addition, there are high interfactor correlations (e.g., .94), which probably indicates that internal structure should be revised using a bifactor and/or bifactor ESEM model. Finally, to make valid comparisons by gender or other variable, it is necessary to perform measurement invariance of the scale. 

Author Response

Reviewer 3 Report (New Reviewer)

Round 2

Reviewer 3 Report (New Reviewer)

10-10-2022

 Dear authors,

Thank you for submitting a revised version of your manuscript The measurement of love: Psychometric properties and prelimi-2 nary findings of the Short Love Scale (SLS-12) in a Polish sample. The first version of your work has a few critical shortcuts that call into question its validity and scientific contributions.

After reading your responses to my comments I feel that the scientific content of your manuscript has reached an acceptable standard for its publication in IJERPH, and there is no need to send it for another revision round. You have successfully dealt with most of the comments about the theoretical framework, methodological issues, and discussion improvement.

I wish you the very best for your work.

Best regards.

This manuscript is a resubmission of an earlier submission. The following is a list of the peer review reports and author responses from that submission.

Round 1

Reviewer 1 Report

This study developed the Short Love Scale (SLS-12) and examined its psychometric qualities in a Polish sample.

Major concerns

1. Lines 164-165, the authors selected 941 participants from the total sample (N=2050). As stated in Tables S1 to S2, the 941 responses were submitted to EFA. Based on the information, I assume the authors conducted EFA and CFA on the same dataset. This practice is methodologically wrong. Data used for EFA cannot be used for CFA.

2. The introduction of the EFA analysis method and results are insufficient:

(a) While the one-factor model was suggested by referring to the eigenvalue, it is unknown how the four-factor model and 3-factor model were identified.

(b) Tables S2 and S3 show that some items loaded on more than one factor. Moreover, the factor loadings are quite similar (e.g., Item 4 in Table S2). This suggests that the item does not clearly belong to one factor. Why were the items kept? How did the authors determine the belonging of the item? 

(c) what is the reason to use varimax when the factors are assumed to be correlated with each other?

3. Both the three-factor theory driven SLS-12 model and three-factor SLS-12 model (based on EFA solution) showed a good fit to the data and their results are similar. Why is the latter preferred?

Minor concerns 

1. The results for the second-order models shall be presented in Table 3.

2. Report the estimator used in CFA.

3. Rename the two three-factor models to ease readers to distinguish them.

Reviewer 2 Report

Thank you very much for the corrections and answers. Congratulations on the study.